# L3MBTL3 Is a Potential Prognostic Biomarker and Correlates with Immune Infiltrations in Gastric Cancer

**DOI:** 10.3390/cancers16010128

**Published:** 2023-12-27

**Authors:** Lin Gan, Changjiang Yang, Long Zhao, Shan Wang, Yingjiang Ye, Zhidong Gao

**Affiliations:** 1Department of Gastroenterological Surgery, The First Affiliated Hospital of Zhengzhou University, Zhengzhou 450052, China; 2011110395@stu.pku.edu.cn; 2Department of Gastroenterological Surgery, Peking University People’s Hospital, Beijing 100044, China; 2211110425@stu.pku.edu.cn (C.Y.); 2111110407@pku.edu.cn (L.Z.); wangshan@pkuph.edu.cn (S.W.); 3Laboratory of Surgical Oncology, Peking University People’s Hospital, Beijing 100044, China

**Keywords:** L3MBTL3, gastric cancer, prognosis, immune infiltration

## Abstract

**Simple Summary:**

Gastric cancer (GC) is a severe disease with high mortality. There is an immediate demand for GC prognostic markers. The biological role of the lethal (3) malignant brain tumor-like 3 (L3MBTL3) has been investigated in human malignancies, but its involvement in GC is not well understood. This research seeks to better understand L3MBTL3’s prognostic significance and expression profiles in GC. We examined the expression and prognostic significance of L3MBTL3 in GC using bioinformatic analysis. A functional enrichment analysis was conducted to explore the potential mechanisms. The correlations between L3MBTL3 and immune infiltration were determined based on the TIMER database. Our work revealed that L3MBTL3 is a promising diagnostic and prognostic marker for GC, opening new avenues for research and treatment.

**Abstract:**

Recent research has linked lethal (3) malignant brain tumor-like 3 (L3MBTL3) to cancer aggressiveness and a dismal prognosis, but its function in gastric cancer (GC) is unclear. This research investigated the association between L3MBTL3 expression and clinicopathological characteristics of GC cases, as well as its prognostic value and biological function based on large-scale databases and clinical samples. The results showed that L3MBTL3 expression was upregulated in malignant GC tissues, which was associated with a shortened survival time and poor clinicopathological characteristics, including TNM staging. A functional enrichment analysis including GO/KEGG and GSEA illustrated the enrichment of different L3MBTL3-associated pathways involved in carcinogenesis and immune response. In addition, the correlations between L3MBTL3 and tumor-infiltrating immune cells were determined based on the TIMER database; the results showed that L3MBTL3 was associated with the immune infiltration of macrophages and their polarization from M1 to M2. Furthermore, our findings suggested a possible function for L3MBTL3 in the regulation of the tumor immune microenvironment of GC. In summary, L3MBTL3 has diagnostic potential, and it also offers new insights into the development of aggressiveness and prognosis in GC.

## 1. Introduction

Gastric cancer (GC) is one of the most prevalent cancers of the digestive tract, and it ranks number four among the leading contributors to cancer-related death, which seriously endangers human health. In 2020, over a million people were diagnosed with GC, and approximately 80,000 deaths occurred as a result of the disease [1]. Regrettably, the diagnosis is often made at a late stage in the disease’s progression, when only palliative therapy options are feasible. The prognosis is not favorable, notwithstanding the progress that has been made in treatment, which includes surgeries, radiation, immunotherapy, and chemotherapy. The overall survival (OS) rate of patients diagnosed with stage IV GC at 5 years is <10% [2]. The early diagnosis and treatment of GC are the two most important factors in improving the patient’s prognosis. Thus, it is essential to conduct research into reliable biological markers that can accurately anticipate the prognosis and identify targets for prospective treatment.

The protein identified as L3MBTL3 (Lethal (3) malignant brain tumor-like 3), which also goes by the name MBT1, is a transcriptional repressor that belongs to the MBT (malignant brain tumor) family and contains MBT domains [3]. L3MBTL3 was previously isolated as a PcG (Polycomb group) protein in mammals that likely facilitates chromatin modification and compaction and is highly expressed in hematopoietic progenitor cells [4]. The null L3MBTL3 mutation in mice caused them to have anemia, which led to early embryonic lethality [5]. L3MBTL3 is composed of three repeated MBT domains, each of which acts as a mono- or demethylated lysine reader for histones [6,7]. L3MBTL3 has been linked to a role in protein breakdown that is mediated by methylation. Methylated DNMT1, E2F, or SOX2 was shown to be bound by L3MBTL3, which then engaged a ubiquitin ligase for their degradation [8,9]. Nonetheless, there is no concrete proof of L3MBTL3 involvement in GC, and its additional biological roles are largely unidentified.

Herein, the expression of L3MBTL3 was initially compared between GC and normal tissues using data retrieved from the Gene Expression Omnibus (GEO) [10] and The Cancer Genome Atlas (TCGA) [11]. Next, to further elucidate the clinical significance of L3MBTL3 in GC patients, we analyzed the association of L3MBTL3 expression patterns with patient clinical characteristics and survival statuses. In addition, we investigated the association of L3MBTL3 mRNA levels with the immune microenvironment of tumors and the possible mechanism behind this association. These findings provide light on the function of L3MBTL3 in GC onset and progression and subsequent immune response.

## 2. Materials and Methods

### 2.1. Collection and Preprocessing of Data

Stomach adenocarcinoma (STAD) [11] RNA-Seq, gene expression, and corresponding clinical data of 375 GC patients and 32 normal tissues were retrieved from the TCGA database [11] (https://portal.gdc.cancer.gov/ (accessed on 7 July 2023)). GSE19826 [12] and GSE65801 [13] were selected and downloaded from Gene Expression Omnibus (GEO) [10] (http://www.ncbi.nlm.nih.gov/geo (accessed on 12 July 2023)).

### 2.2. Patients and Clinical Specimens

Our research was approved by the ethical committees of the Peking University People’s Hospital and was conducted as guided by the Declaration of Helsinki. Then, informed written consent was provided by all patients. The participants included 40 eligible patients who underwent surgical treatment for GC at the hospital between May 2022 and January 2023. None of the patients had previously been treated with radiotherapy or chemotherapy before having the surgery. All of the patients with GC had their diagnoses confirmed via pathological evaluation. For the immunohistochemical (IHC) analysis, we fixed tumor tissues together with adjacent normal tissues, which were surgically excised and located at a distance >5 cm from the tumor, in a 4% formalin solution and subsequently embedded them in paraffin blocks.

### 2.3. Clinical Characteristics and Survival Analysis

The link between L3MBTL3 expression and age, clinical stage, and TNM stage, as well as other clinical features, was investigated by means of Pearson’s chi-square test or Wilcoxon signed-rank test, as appropriate. Pan-cancer survival was analyzed using a gene expression profiling interactive analysis (GEPIA [14]: http://gepia2.cancer-pku.cn/ (accessed on 15 July 2023)). The survival analysis was conducted via the Kaplan–Meier (KM) technique and the log-rank test, using the median of L3MBTL3 expression as the cutoff value. The Kaplan–Meier plotter tool [15,16] (https://kmplot.com/analysis/ (accessed on 17 July 2023)) was also used to measure the patient survival rate according to L3MBTL3 expression.

### 2.4. Enrichment Analysis

The biological involvement of L3MBTL31 in GC was investigated by comparing high- and low-expression groups via differential gene expression analyses (DEGs), with |logFC| > 1 and FDR < 0.05 as the parameters for significant DEGs. Thereafter, the ClusterProfiler package in R (3.6.3) was utilized to undertake Gene Ontology (GO) [17], Kyoto Encyclopedia of Genes and Genomes (KEGG) enrichment analysis [18], and gene set enrichment analysis (GSEA) [19] to find enriched molecular mechanisms and cellular functions. The GO analysis includes cellular components (CCs), molecular functions (MFs), and biological processes (BPs). GSEA is a computation technique for examining if an a priori-determined collection of genes has concordant variations in two biological states and statistical significance [19]. To further organize the enriched pathways, encompassing KEGG and hallmark pathways, for each phenotype, we employed the adjusted *p*-value and normalized enrichment score (NES). Gene sets with a false discovery rate (FDR) < 0.25 and adjusted *p* < 0.05 were considered significantly enriched.

### 2.5. Immune Infiltration Analysis

L3MBTL3 was also correlated with the infiltration level of immune cells, such as dendritic cells (DCs), macrophages, B cells, neutrophils, and T cells (both CD4+ and CD8+), using data from the Tumor Immune Estimation Resource (TIMER) database [20,21]. Spearman’s correlation analysis was implemented to examine whether L3MBTL3 expression was linked to these immune cells. Additionally, a heat map chart was used to visualize the association between immune modulators and L3MBTL3 expression. Using Spearman’s correlation analysis, it was determined if the levels of L3MBTL3 and these immune factors were correlated.

### 2.6. Immunohistochemistry

To determine if L3MBTL3 was expressed differently in normal tissues compared with malignant tissues, IHC was used. We used dimethyl benzene to dewax and gradient ethanol solutions to rehydrate paraffin-embedded sections of gastric cancer and adjoining normal tissue. Antigen retrieval with the sodium citrate solution was performed by microwaving at 95 °C. The endogenous peroxidase was then blocked by using 3% H_2_O_2_ for 10 min. The next step involved blocking and incubating the tissue slices in 10% fetal bovine serum (FBS) for 1 h. The primary antibody L3MBTL3 (1:200; 28112-1-AP) was applied and left to incubate overnight at 4 °C. The subsequent incubation process employed a dark environment and involved the utilization of an HRP-labeled universal anti-rabbit secondary antibody. Immunostaining outcomes were visualized using diaminobenzidine, and the tissue sections experienced counterstaining with hematoxylin. Following this, the sections underwent dehydration, mounting, and examination for the expression of L3MBTL3 by two pathologists (HM Zhao and RR Luan). These professionals assessed the expression level by considering both the proportion of cells testing positive and the intensity of the staining. The categorization of positive cells was as follows: 0 (0–10%), 1 (10–40%), 2 (40–70%), or 3 (>70%), while the staining intensity was rated on a scale of 1 (weak), 2 (moderate), or 3 (strong). The final L3MBTL3 IHC scores were determined by combining the proportion of cells exhibiting positive staining and the staining intensity. 

### 2.7. Statistical Analysis

We employed the statistical software R version 3.6.3 to perform the analysis of the TCGA data. The Wilcoxon rank-sum test and the Wilcoxon signed-rank test were utilized for comparing the expression of L3MBTL3 between the tumor and control groups. To assess the relationship between L3MBTL3 expression and clinicopathological parameters, we utilized Welch’s one-way ANOVA followed by Bonferroni’s post hoc test (or *t*-test). To explore how various clinical characteristics influenced the expression of L3MBTL3, we conducted Pearson’s chi-square test. The prognostic significance of L3MBTL3 was evaluated using the KM technique to analyze overall survival (OS), disease-specific survival (DSS), and progression-free interval (PFI). To determine the ability of L3MBTL3 to differentiate, a receiver operating characteristic (ROC) analysis was conducted using the pROC package. The calculated area under the curve (AUC) values ranged from 0.5 to 1.0, indicating a discrimination ability between 50% and 100%. We considered statistical significance when the two-tailed *p*-value was ≤0.05. 

## 3. Results

### 3.1. The Expression of L3MBTL3 in GC

In this study, we utilized data from the TCGA database to investigate the gene expression levels of L3MBTL3 in various human cancer tissues compared with normal tissues. Our analysis of unpaired samples revealed that mRNA expression of L3MBTL3 was significantly higher in multiple types of cancer tissues, such as cholangiocellular cancer (CHOL), colon cancer (COAD), esophageal cancer (ESCA), head and neck squamous cell cancer (HNSC), lung cancer (LUAD and LUSC), prostate cancer (PRAD), and GC (STAD). However, L3MBTL3 expression was downregulated in breast cancer (BRCA), cervical cancer (CESC), kidney chromophobe (KICH), Pheochromocytoma and Paraganglioma (PCPG), thyroid cancer (THCA), and uterine corpus endometrial carcinoma (UCEC) (Figure 1A). To obtain a more accurate reflection of L3MBTL3 expression changes in tumors, we also conducted a comparative analysis of paired samples from the TCGA database, which yielded similar results overall. However, pairwise analysis showed expression differences in bladder cancer (BLCA), while cervical cancer (CESC), colon cancer (COAD), lung squamous cell carcinoma (LUSC), and Pheochromocytoma and Paraganglioma (PCPG) did not show significant differences (Figure 1B). L3MBTL3 expression was found to be upregulated or downregulated in different tumors, suggesting that it may have varying roles in different tumor types. A survival analysis in pan-cancer also revealed that the expression of L3MBTL3 has different or even opposite effects on the prognosis of different tumors. For instance, cervical cancer (CESC), lung cancer (LUAD), and GC (STAD) with high expression of L3MBTL3 were associated with poorer prognosis, while the opposite was observed in cholangiocellular cancer (CHOL) and kidney cancer (KIRC) (Figure 1C). In GC, high expression of L3MBTL3 was significantly associated with an unfavorable prognosis, affecting both OS and disease-free survival (DFS) (Figure 1C). Further analysis of unpaired and paired samples from the TCGA database confirmed significant upregulation of L3MBTL3 in GC (Figure 1D,E). Additionally, the upregulation of L3MBTL3 expression in tumor tissues compared with normal tissues was also observed in two other independent datasets: GSE19826 and GSE65801 (Figure 1F). To further validate these findings, immunohistochemistry (IHC) staining was performed on 40 paired cancer and non-cancer tissues from gastric cancer patients at Peking University People’s Hospital. Consistent with the previous results, the IHC scores for L3MBTL3 were significantly higher in tumor tissues compared with normal tissues (Figure 1G,H). The ROC curve analysis of the TCGA data demonstrated that L3MBTL3 expression had a remarkable predictive capacity to distinguish cancer tissues from non-cancer tissues, with an AUC of 0.805 (95% CI = 0.738–0.872) (Figure 1I). In summary, the analysis conducted in this study provides strong evidence that L3MBTL3 expression is significantly upregulated in GC.

### 3.2. Association of L3MBTL3 Expression with Clinical Features and Prognostic Value of L3MBTL3 in GC

We next evaluated the 362 patients in the TCGA database for associations between L3MBTL3 expression and clinicopathological characteristics. L3MBTL3 expression was shown to be strongly associated with T stage, N stage, pathological stage, histologic grade, OS, DSS, and PFI event using Welch one-way ANOVA with Bonferroni correction (Figure 2). Meanwhile, the median L3MBTL3 expression level was employed to characterize the expression data into high- and low-expression groups. The clinicopathologic characteristics of sex, age at the diagnosis, T stage, N stage, clinical stage, histologic grade, and distant metastasis were then correlated with L3MBTL3 expression in cancer tissues. Table 1 illustrates the association of L3MBTL3 expression with the clinical characteristics of GC patients. L3MTBL3 overexpression was linked to a higher T stage (*p* = 0.009), N stage (*p* = 0.002), and histologic grade (*p* = 0.025) compared with low expression. There was no significant relationship between L3MBTL3 expression and any other clinicopathologic variables. As for the IHC cohort, the overexpression of L3MBTL3 was linked to advanced T stage (*p* = 0.035), N stage (*p* = 0.007), and clinical stage (*p* = 0.032) compared with low expression (Table 2).

We analyzed the prognosis-predictive power of L3MBTL3 for OS, DSS, and PFI in a comprehensive set of GC cases. Patients with lower L3MBTL3 expression exhibited longer OS, DSS, and PFI, according to KM survival analysis (Figure 3A–C). Kaplan–Meier Plotter [16] was used to validate L3MBTL3’s prognostic significance (Figure 3D,E).

### 3.3. Functional Inference of L3MBTL3 in GC

Overall, 500 DEGs were found between high- and low-L3MBTL3-expression groups, with 404 genes showing upregulation and 126 showing downregulation in the high-expression group (Figure 4A). We also carried out GO and KEGG pathway analyses. Epidermis development, the intermediate filament cytoskeleton, skin epidermis structural components, and pancreatic secretion were revealed to be overrepresented among the downregulated genes (Figure 4D). On the other hand, the upregulated genes were found to be primarily enriched in collagen-containing extracellular matrix, cell–cell adhesion via plasma membrane adhesion molecules, signaling receptor activator activity, neuroactive ligand–receptor interaction, the PI3K-AKT signaling pathway, etc. (Figure 4E). Moreover, GSEA revealed that pathways associated with ECM receptor interaction, the Hedgehog signaling pathway, cell adhesion molecules cams, the calcium signaling pathway, and focal adhesion were enriched in patients exhibiting L3MBTL3 upregulation, while low L3MBTL3 expression was associated with the immune response including the immunoglobulin complex, immunoglobulin receptor binding, antigen binding, and circulatory immunoglobulin-mediated humoral immune response (Figure 4B,C). Altogether, these gene enrichment studies illustrate that L3MBTL3 is important in the immune response of GC, as well as the invasion of cancer cells via cell adhesion pathways.

### 3.4. L3MBTL3 Expression Correlates with Infiltration Levels of Macrophages

The TIMER database was employed to study the association of L3MBTL3 expression with several types of infiltrating immune cell levels including DCs, macrophages, neutrophils, T cells (CD4+ and CD8+), and B cells. Notably, L3MBTL3 expression was most strongly linked to macrophage infiltration (r = 0.402, *p* < 0.001), which is a key constituent of the tumor microenvironment (TME) and plays a role in regulating the process of angiogenesis, the proliferation of malignant cells, metastasis, extracellular matrix remodeling, and immunosuppression, along with checkpoint blockade immunotherapy and resistance to chemotherapeutic agents (Figure 5A–F) [22]. Furthermore, we compared the correlation between the L3MBTL3 expression and different subtypes of macrophages, including M1 and M2 through CIBERSORT and CIBERSORT-ABS based on TIMER. L3MBTL3 was shown to have a strong association with M2 but not with M1 macrophages (Figure 5G–J). We also examined how L3MBTL3 expression related to macrophage markers. L3MBTL3 was shown to have a much stronger relationship with M2 macrophages (CD163, r = 0.257, *p* < 0.001, MRC1: r = 0.281, *p* < 0.001) than with CD68 (r = 0.018, *p* = 0.733) (Figure 6A–C). These data illustrate a putative role of L3MBTL3 in M2 polarization in GC.

Considering that cancer cells may enhance immune cell polarization through chemokines and chemokine receptors, the present research established a correlation between the expression of L3MBTL3 and chemokines and receptors based on the TISIDB database (Figure 6D–F). Notably, CXCL12 (r = 0.301, *p* < 0.001) and CXCR4 (r = 0.328, *p* < 0.001) were identified as the most important chemokine and receptor, respectively (Figure 6G). Intriguingly, patients with low CXCL12 and CXCR4 expression exhibited superior overall survival outcomes compared with those with high expression, similar to the findings for L3MBTL3 (Figure 6H,I). Moreover, CXCL12 and CXCR4 were also strongly correlated with macrophages and the markers of M2 macrophages (Figure 6J–L).

## 4. Discussion

As the fourth leading contributor to cancer-related mortality, gastric cancer is among the most prevalent forms of the disease worldwide [1]. STAD is responsible for 95% of all occurrences of gastric cancer, as determined by the pathological staging of the disease [23]. The dismal survival rates of patients with GC are likely caused by numerous confounding variables. Patients diagnosed at an early stage with disease limited to the mucosa and submucosa and who undergo surgical treatment have better five-year survival rates, while patients with advanced stages are always accompanied by recurrence and metastasis after an operation, which is the most common reason for the poor prognosis [24]. Consequently, it is crucial to formulate reliable biological markers for the early detection of GC and the provision of appropriate therapy targets to improve patient prognoses.

The malignant brain tumor (MBT) family is a large family with L3MBTL1, L3MBTL2, L3MBTL3, L3MBTL4, and other members. L3MBTL3 is a gene with 30 exons that may be found on the long arm of chromosome 6. In MBT, it interacts with chromatin and belongs to a class of transcriptional inhibitors [6]. Because of its ability to bind to methyl lysine in histone, it has a multiplier effect on the aberrant transcription regulation that is seen in many disorders. A polymorphism in the L3MBTL3 gene was shown to be linked to a greater risk of multiple sclerosis [25]. The development and maturation of myeloid progenitor cells were shown to be uniquely controlled by MBT-1, according to research by Satoko Arai et al. Additionally, by temporarily elevating p57KIP2 expression, MBT-1 was also shown to influence bone marrow cell formation [5]. Recently, accumulating evidence has revealed that L3MBTL3 may be a driving force for carcinogenesis. Toth et al. reported that L3MBTL3 mutations were linked to a greater risk of developing multiple types of cancers, including colorectal, overall breast, estrogen receptor (ER)-negative breast, clear cell ovarian, and overall and aggressive prostate cancer risk [26,27]. However, the association and potential mechanism of L3MBTL3 involved in gastric cancer have not been reported.

To facilitate our analysis, we determined the relative abundance of L3MBTL3 mRNA in a TCGA pan-cancer sample. Notably, we discovered that L3MBTL3 mRNA levels were remarkably higher in GC, as well as many other cancers, as opposed to non-cancer tissues. The elevated L3MBTL3 levels in GC were additionally corroborated in separate GEO datasets and the IHC cohort from our hospital. In particular, L3MBTL3 overexpression was linked to a dismal survival rate, including OS, DSS, and PFI. Moreover, an elevated L3MBTL3 level was substantially linked to the advanced stage of GC patients. Results like these highlighted the possible function of L3MBTL3 as a predictive and diagnostic marker in GC tumor growth.

Additionally, according to the GO/KEGG enrichment, the potential mechanism of L3MBTL3 might involve the collagen-containing extracellular matrix, neuroactive ligand-receptor interaction, plasma membrane adhesion molecule-mediated cell–cell adhesion, signaling receptor activator activity, the PI3K-AKT signaling pathway, etc. Interestingly, GSEA revealed that pathways associated with the Hedgehog signaling pathway, cell adhesion molecule cams, the calcium signaling pathway, focal adhesion, and ECM receptor interaction were enriched in the high-L3MBTL3-expression group, while low L3MBTL3 expression was associated with the immune response including the immunoglobulin complex, immunoglobulin receptor binding, antigen binding, and the circulatory immunoglobulin-mediated humoral immune response. These results highlighted that suppressive immune responses were also involved in the L3MBTL3 expression phenotype, indicating that L3MBTL3 might be involved in the immune microenvironment.

Tumor onset and progression rely primarily on the immune microenvironment [28]. Components of the complex TME are required for tumor cell proliferation and spread at every stage of carcinogenesis. Cancer patients who have an imbalanced ratio of immune cell components have a much worse prognosis [29]. Furthermore, the TME immune cells may be employed for prognostic evaluation of different types of malignancies [30]. The tumor immune microenvironment (TIME) is now considered an important factor during GC development. In our study, we evaluated the L3MBTL3 expression and GC TME. The study suggested that L3MBTL3 expression was closely related to macrophages. Macrophages have a crucial function in maintaining innate immunity, tissue homeostatic balance, and inflammatory processes, making them a key component of the tumor immunosuppressive microenvironment [31,32]. By undergoing a process of polarization, macrophages may transition from a classically activated M1 (proinflammatory) condition to an alternatively activated M2 (anti-inflammatory) one [33]. Anti-tumor immunity may be boosted by M1 macrophages. M2 macrophages, on the other hand, are pro-tumor since they either directly promote angiogenesis or indirectly induce immune suppression [34]. L3MBTL3 was strongly linked to M2 macrophages, as shown by our analysis. Based on these findings, L3MBTL3 may enhance GC onset and progression by activating M2 macrophages (Figure 7).

We examined the links between L3MBTL3 expression and chemokines and chemokine receptors to understand more about the mechanism via which L3MBTL3 regulates the immune microenvironment. Our attention was particularly drawn to the chemokine CXCL12 and its receptor CXCR4 due to their notable correlation with L3MBTL3. According to the report, the binding of CXCL12 to CXCR4 resulted in the activation of signal transduction pathways that had extensive impacts on chemotaxis, cell proliferation, migration, and gene expression. Consequently, this axis functioned as a conduit for communication between tumor and stromal cells, thereby establishing a conducive microenvironment for tumor survival, development, angiogenesis, and metastasis [35]. Additionally, the CXCL12–CXCR4 axis played a role in the recruitment of M2-polarized macrophages, which corroborated our prior findings [36,37].

Obviously, L3MBTL3 has a biological role in boosting GC oncogenesis. However, some limitations need to be considered. First, the main data of our study were mainly collected from public databases, including TCGA and GEO. Only 40 tissue samples from our center were used for validation, which is a small sample size. Additional clinical studies with more centers, larger sample sizes, and more dimensions (genome, transcriptome, and proteome) are required to further verify the relationship between L3MBTL3 expression and GC. Second, we need to carry out further in vitro and in vivo experimental research on cell lines and animal models to comprehensively validate the biological functions of L3MBTL3 and the fundamental modulatory mechanisms in GC onset and advancement and to also reveal how L3MBTL3 is involved in the regulation of the tumor immune microenvironment and immune infiltration of GC. We will remedy these shortcomings in our future research.

## 5. Conclusions

In summary, our research shows that L3MBTL3 is extensively expressed in GC and that upregulated expression levels correlate strongly with unfavorable survival outcomes and M2 macrophages. Therefore, L3MBTL3 may be a prognostic marker for GC patients.

## Figures and Tables

**Figure 1 cancers-16-00128-f001:**
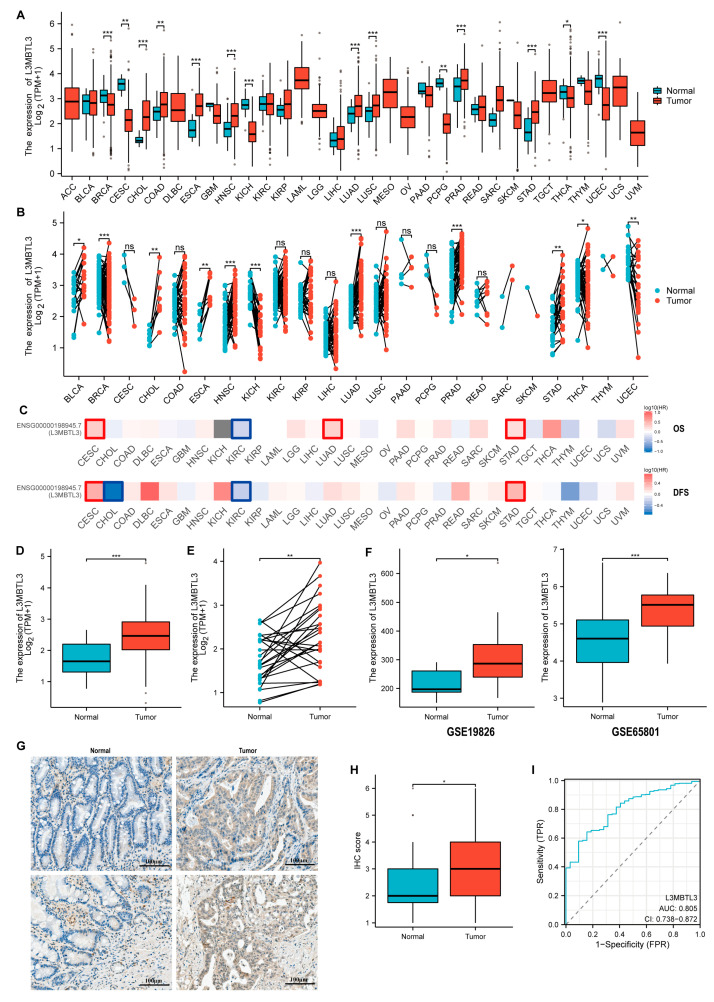
The expression levels of L3MBTL3 in pan-cancer and GC tissues compared with normal samples. (**A**) Comparative analysis of unpaired samples shows that L3MBTL3 mRNA expression was upregulated or downregulated in different tumor tissues versus normal tissues based on the TCGA database. (**B**) Comparative analysis of paired samples shows that L3MBTL3 mRNA expression was upregulated or downregulated in different tumor tissues versus normal tissues based on the TCGA database. (**C**) L3MBTL3 had different prognostic effects on different tumors and was a poor prognostic factor for GC. Red represents risk factors, while blue represents protective factors, and square borders indicate statistically significant differences. (**D**,**E**) Comparative analyses of unpaired and paired samples both show that L3MBTL3 mRNA expression was upregulated in GC tissues versus normal tissues based on the TCGA database. (**F**) L3MBTL3 mRNA expression was upregulated in GC tissues versus normal tissues based on the GEO database. (**G**) IHC staining showed that L3MBTL3 protein expression was upregulated in GC tissues compared with normal tissues. (**H**) The IHC score of GC tissues was higher than that of normal tissues. (**I**) ROC analysis shows that L3MBTL3 had good diagnostic efficacy for GC. (Abbreviations: GC, gastric cancer; L3MBTL3, Lethal (3) Malignant Brain Tumor-like 3; TPM, transcripts per million; ACC, adrenocortical carcinoma; BLCA, bladder urothelial carcinoma; BRCA: breast carcinoma; CESC, cervical squamous cell carcinoma and endocervical adenocarcinoma; CHOL, cholangiocarcinoma; COAD, colon adenocarcinoma; DLBC, diffuse large B-cell lymphoma; ESCA, esophageal carcinoma; GBM, glioblastoma multiforme; HNSC, head and neck squamous cell carcinoma; KICH, kidney chromophobe; KIRC, kidney renal clear cell carcinoma; KIRP, kidney renal papillary cell carcinoma; LAML, acute myeloid leukemia; LGG, brain lower-grade glioma; LIHC, liver hepatocellular carcinoma; LUAD, lung adenocarcinoma; LUSC, lung squamous cell carcinoma; MESO, mesothelioma; OV, ovarian serous cystadenocarcinoma; PAAD, pancreatic adenocarcinoma; PCPG, pheochromocytoma and paraganglioma; PRAD, prostate adenocarcinoma; READ, rectum adenocarcinoma; SARC, sarcoma; SKCM, skin cutaneous melanoma; STAD, stomach adenocarcinoma; TGCT, testicular germ cell tumors; THCA, thyroid carcinoma; THYM, thymoma; UCEC, uterine corpus endometrial carcinoma; UCS, uterine carcinosarcoma; UVM, uveal melanoma; OS, overall survival; DFS, disease-free survival; IHC, immunohistochemistry; TPR, true positive rate; FPR, false positive rate; AUC, area under curve; CI, confidence interval; TCGA, the Cancer Genome Atlas; GEO, Gene Expression Omnibus; ns, no significance; * *p* < 0.05; ** *p* < 0.01; *** *p* < 0.001).

**Figure 2 cancers-16-00128-f002:**
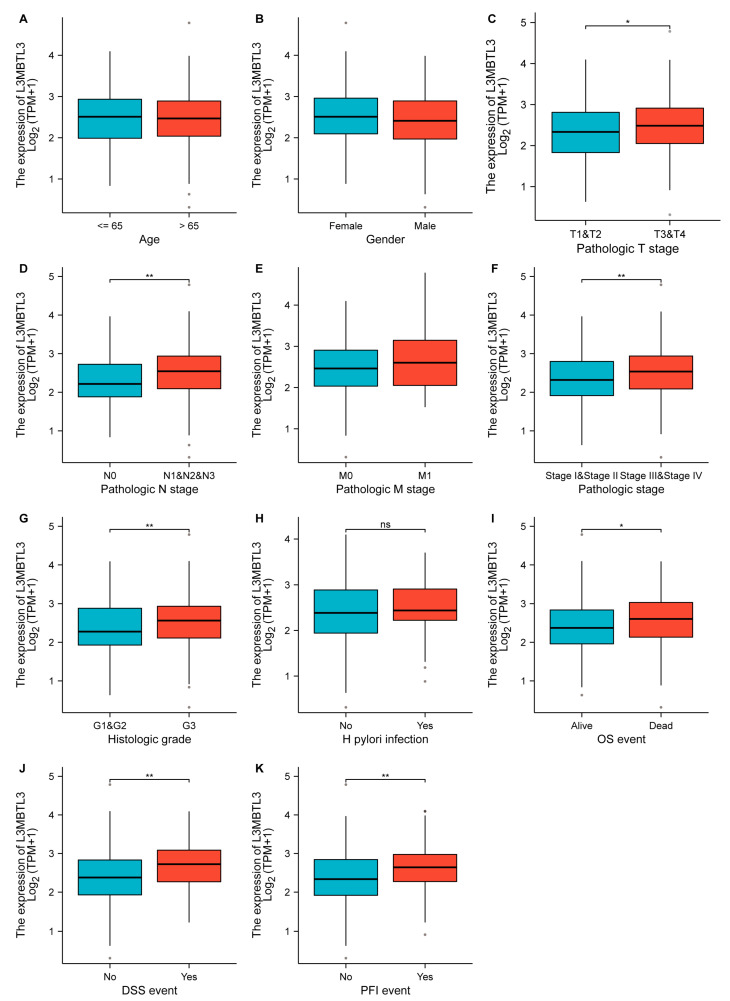
Correlation between L3MBTL3 expression and clinicopathological characteristics of GC based on TCGA database. (**A**) Age. (**B**) Gender. (**C**) Pathologic T stage. (**D**) Pathologic N stage. (**E**) Pathologic M stage. (**F**) Pathological stage. (**G**) Histologic grade. (**H**) H. pylori infection. (**I**) OS event. (**J**) DSS event. (**K**) PFI event. (Abbreviations: GC, gastric cancer; L3MBTL3, Lethal (3) Malignant Brain Tumor-like 3; TPM, transcripts per million; T, tumor; N, node; M, metastasis; G, grade; OS, overall survival; DSS, disease-specific survival; PFI, progress free interval; ns, no significance; * *p* < 0.05; ** *p* < 0.01.)Association between L3MBTL3 expression and clinicopathological characteristics of GC based on TCGA database. (**A**) Age. (**B**) Sex. (**C**) T stage. (**D**) N stage. (**E**) M stage. (**F**) Pathological stage. (**G**) Histologic grade. (**H**) H. pylori infection. (**I**) OS event. (**J**) DSS event. (**K**) PFI event. (Abbreviations: GC, gastric cancer; L3MBTL3, Lethal (3) Malignant Brain Tumor-like 3; TPM, transcripts per million; T, tumor; N, node; M, metastasis; G, grade; OS, overall survival; DSS, disease-specific survival; PFI, progress free interval; ns, no significance; * *p* < 0.05; ** *p* < 0.01).

**Figure 3 cancers-16-00128-f003:**
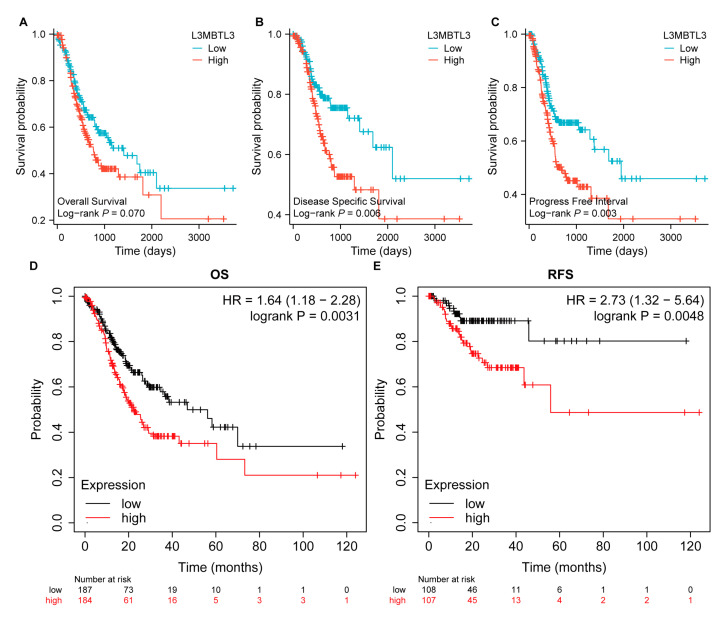
Analysis of the prognostic significance of L3MBTL3 in GC. (**A**–**C**) The Kaplan–Meier curves show that GC patients with higher expression of L3MBTL3 had a shorter overall survival time, disease-specific survival time, and progress-free interval time based on the TCGA database. (**D**,**E**) Kaplan–Meier Plotter analysis revealed that high expression of L3MBTL3 was associated with a poor OS and RFS of GC patients. (Abbreviations: GC, gastric cancer; L3MBTL3, Lethal (3) Malignant Brain Tumor-like 3; OS, overall survival; RFS, relapse free survival; HR, hazard ratio).

**Figure 4 cancers-16-00128-f004:**
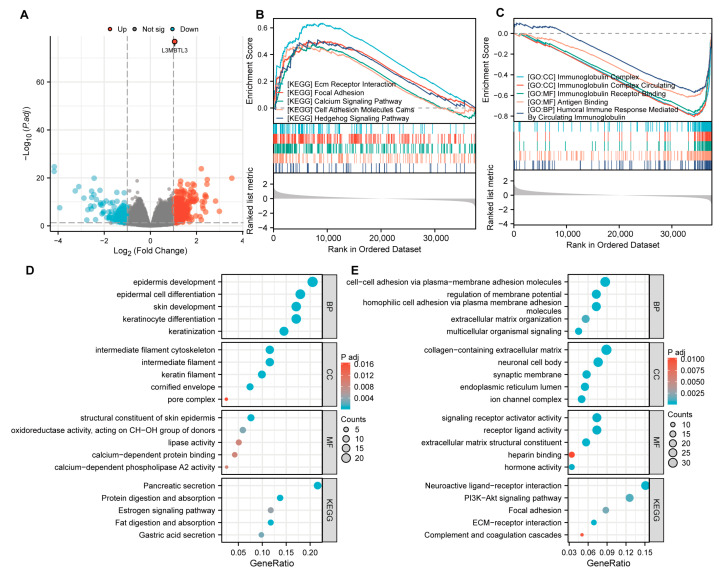
Analysis of DEGs between the high- and low-expression groups of L3MBTL3 of GC patients and subsequent GSEA, GO, and KEGG enrichment analysis. (**A**) Volcano map illustrating DEGs between the high- and low-expression groups of L3MBTL3 in GC. Red represents high expression, and blue represents low expression. GSEA enrichment plots showed positive correlations of high (**B**) and low (**C**) L3MBTL3 expression with different tumor-related pathways. The gray dotted line indicates that the enrichment score is 0, while positive and negative enrichment scores indicate that the gene set is enriched in the high expression group and the low expression group, respectively. GO and KEGG pathway analyses of downregulated DEGs (**D**) and upregulated DEGs (**E**) with a correlation coefficient >0.6 in GC. (Abbreviations: GC, gastric cancer; L3MBTL3, Lethal (3) Malignant Brain Tumor-like 3; DEGs, differential expressed genes; KEGG, Kyoto Encyclopedia of Genes and Genomes; GO, Gene Ontology; CC, cell component; MF, molecular function; BP, biological process).

**Figure 5 cancers-16-00128-f005:**
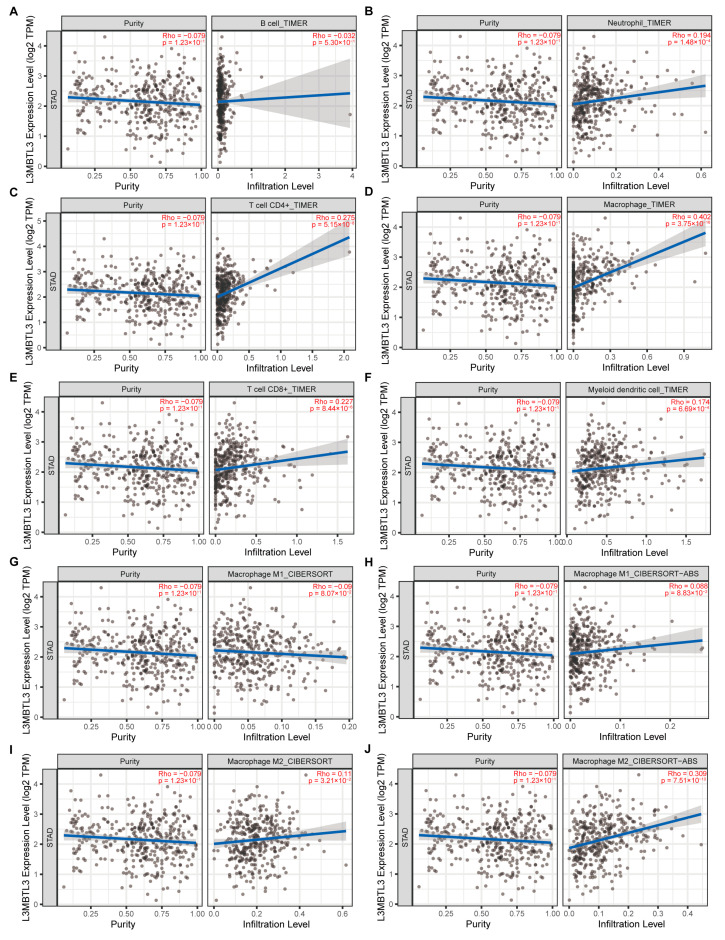
The correlation of L3MBTL3 expression with immune infiltration level in GC. The correlation of L3MBTL3 expression with (**A**) B cell, (**B**) neutrophil, (**C**) CD4+ T cell, (**D**) macrophage, (**E**) CD8+ T cell, and (**F**) myeloid dendritic cell infiltration in GC acquired from the TIMER online tool. L3MBTL3 has a strong association with M2 but not with M1 macrophages (**G**–**J**) through CIBERSORT and CIBERSORT-ABS based on TIMER. (Abbreviations: GC, gastric cancer; L3MBTL3, Lethal (3) Malignant Brain Tumor-like 3; TPM, transcripts per million; Rho, Spearman’s rank correlation coefficient; TIMER, Tumor Immune Estimation Resource.)

**Figure 6 cancers-16-00128-f006:**
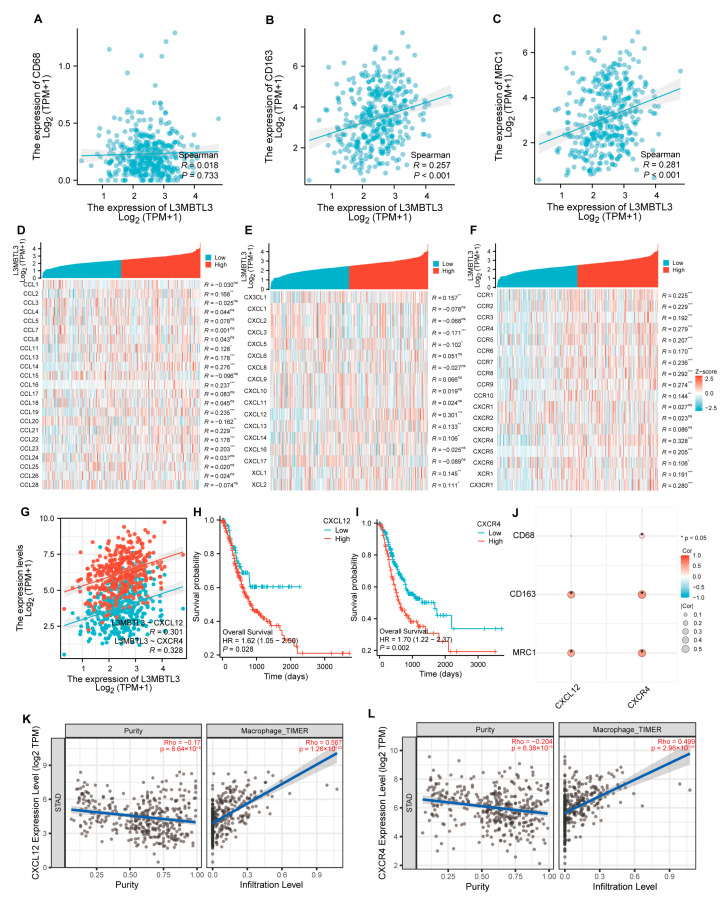
Correlation between L3MBTL3 expression level and macrophage markers, as well as immunological microenvironment’s biomarkers in GC. Correlation between L3MBTL3 expression and macrophage markers CD68 (**A**), CD163 (**B**), and MRC1 (**C**). (**D**,**E**) Correlation between L3MBTL3 and chemokine expression in GC. (**F**) Correlation between L3MBTL3 and chemokine receptor expression in GC. (**G**) Correlation between L3MBTL3 expression and CXCL12 and CXCR4 in GC. (**H**,**I**) Elevated CXCL12 and CXCR4 expression are risk factors for poor overall survival in GC. (**J**) Correlation between CXCL12 and CXCR4 expression and macrophage markers CD68, CD163, and MRC1. CXCL12 (**K**) and CXCR4 (**L**) have a strong association with M2 based on TIMER. (Abbreviations: GC, gastric cancer; L3MBTL3, Lethal (3) Malignant Brain Tumor-like 3; TPM, transcripts per million; R/Rho, Spearman’s rank correlation coefficient; CD, cluster of differentiation; MRC1, Macrophage mannose receptor 1; CCL, Chemokine C-C Motif Ligand; CX3CL1, C-X3-C Motif Chemokine Ligand 1; CXCL, C-X-C Motif Chemokine Ligand; XCL, X-C Motif Chemokine Ligand; CCR, C-C Motif Chemokine Receptor; CXCR, C-X-C Motif Chemokine Receptor; XCR, X-C Motif Chemokine Receptor; CX3CR1, C-X3-C Motif Chemokine Receptor 1; HR, hazard ratio; TIMER, Tumor Immune Estimation Resource; ns, no significance; * *p* < 0.05; ** *p* < 0.01; *** *p* < 0.001).

**Figure 7 cancers-16-00128-f007:**
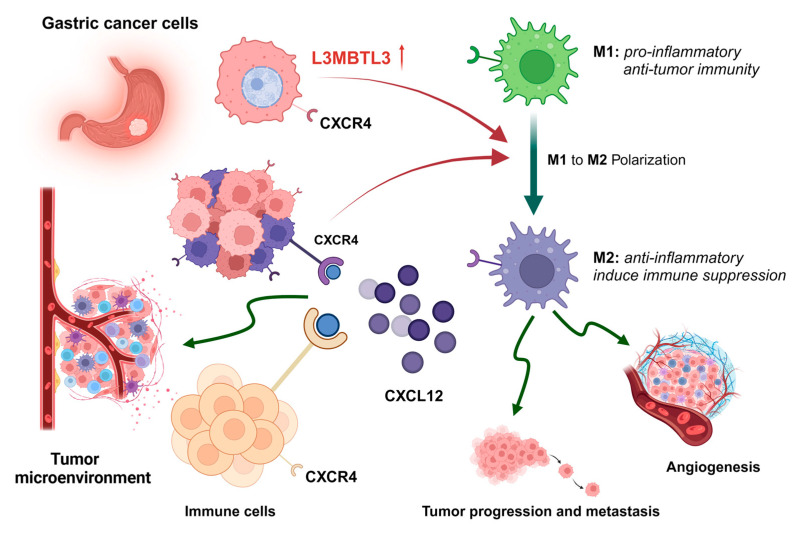
Schematic diagram depicting that upregulated L3MBTL3 in cancer regulates macrophage polarization and the tumor immune microenvironment, ultimately promoting gastric cancer progression. The red arrow indicates that elevated L3MBTL3 can promote polarization from M1 to M2 (thick green arrow). This process can also be enhanced through the CXCL12-CXCR4 interaction, which can also be caused by increased L3MBTL3. The green curved arrow indicates the ultimate biological effects of elevated L3MBTL3, including changes in the tumor immune microenvironment, tumor angiogenesis, tumor progression and metastasis. (Abbreviations: L3MBTL3, Lethal (3) Malignant Brain Tumor-like 3; CXCL12, C-X-C Motif Chemokine Ligand 12; CXCR4, C-X-C Motif Chemokine Receptor 4; M, macrophage.)

**Table 1 cancers-16-00128-t001:** The correlation of clinicopathological characteristics and L3MBTL3 expression in TCGA cohort.

Characteristics	Low Expression of L3MBTL3	High Expression of L3MBTL3	*p* Value
*n*	187	188	
Pathologic T stage, *n* (%)			0.009
T1	15 (4.1%)	4 (1.1%)	
T2	40 (10.9%)	40 (10.9%)	
T3	91 (24.8%)	77 (21%)	
T4	40 (10.9%)	60 (16.3%)	
Pathologic N stage, *n* (%)			0.002
N0	72 (20.2%)	39 (10.9%)	
N1	43 (12%)	54 (15.1%)	
N2	37 (10.4%)	38 (10.6%)	
N3	29 (8.1%)	45 (12.6%)	
Pathologic M stage, *n* (%)			0.335
M0	165 (46.5%)	165 (46.5%)	
M1	10 (2.8%)	15 (4.2%)	
Pathologic stage, n (%)			0.082
Stage I	35 (9.9%)	18 (5.1%)	
Stage II	59 (16.8%)	52 (14.8%)	
Stage III	70 (19.9%)	80 (22.7%)	
Stage IV	17 (4.8%)	21 (6%)	
Sex, n (%)			0.210
Female	61 (16.3%)	73 (19.5%)	
Male	126 (33.6%)	115 (30.7%)	
Age, *n* (%)			0.982
≤65	81 (21.8%)	83 (22.4%)	
>65	102 (27.5%)	105 (28.3%)	
Histologic grade, n (%)			0.025
G1	5 (1.4%)	5 (1.4%)	
G2	81 (22.1%)	56 (15.3%)	
G3	97 (26.5%)	122 (33.3%)	

Abbreviations: L3MBTL3, Lethal (3) Malignant Brain Tumor-like 3; TCGA, the Cancer Genome Atlas; n, number; T, tumor; N, node; M, metastasis; G, grade.

**Table 2 cancers-16-00128-t002:** The correlation of clinicopathological characteristics and L3MBTL3 expression in IHC cohort.

Characteristics	Low Expression of L3MBTL3	High Expression of L3MBTL3	*p* Value
*n*	21	19	
Sex, *n* (%)			0.905
Male	19 (47.5%)	16 (40%)	
Female	2 (5%)	3 (7.5%)	
Age, mean ± sd	60.238 ± 10.222	60.158 ± 15.833	0.985
Pathologic T stage, *n* (%)			0.035
T3 and T4	13 (32.5%)	18 (45%)	
T1 and T2	8 (20%)	1 (2.5%)	
Pathologic N stage, *n* (%)			0.007
N0 and N1	12 (30%)	3 (7.5%)	
N2 and N3	9 (22.5%)	16 (40%)	
Pathologic M stage, *n* (%)			0.196
M0	21 (52.5%)	16 (40%)	
M1	0 (0%)	3 (7.5%)	
Pathologic stage, n (%)			0.032
Stage III and stage IV	11 (27.5%)	16 (40%)	
Stage I and stage II	10 (25%)	3 (7.5%)	

Abbreviations: L3MBTL3, Lethal (3) Malignant Brain Tumor-like 3; IHC, immunohistochemistry; n, number; T, tumor; N, node; M, metastasis.

## Data Availability

The data in this study mainly came from public databases, including The Cancer Genome Atlas (TCGA) database, https://portal.gdc.cancer.gov/ (accessed on 7 July 2023); The Gene Expression Omnibus (GEO) database, http://www.ncbi.nlm.nih.gov/geo (accessed on 12 July 2023); and The Tumor Immune Estimation Resource (TIMER), http://cistrome.shinyapps.io/timer (accessed on 19 July 2023). The data of cases from our hospital cannot be disclosed temporarily.

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
