# Peer review of "L3MBTL3 Is a Potential Prognostic Biomarker and Correlates with Immune Infiltrations in Gastric Cancer"

_cancers, 2023, doi:10.3390/cancers16010128_

Round 1

Reviewer 1 Report

Comments and Suggestions for Authors

            The manuscript entitled “L3MBTL3 Is a Potential Prognostic Biomarker and Correlates with Immune Infiltrations in Gastric Cancer” by Gan et al. is mainly centred in an in silico analysis of data downloaded from publicly available databases on cancer and shows how the L3MBTL3 gene “may be a prognostic marker for gastric cancer patients”. The manuscript is well written and the results presented substantiate that conclusion. Nevertheless, many of the presented results are also available in some web sites (e.g., GEPIA) and the only experimental results presented describe an immunohistochemical analysis of the expression of the gene in tumour and surrounding normal tissues in a local cohort of 40 gastric cancer patients. The absence of more experiments to validate the in silico approaches, to expand them and to confirm in that way the conclusions is a serious objection to the manuscript.

Comments on the Quality of English Language

The quality of English is, generally speaking, fine. Only minor corrections would be needed.

Reviewer 2 Report

Comments and Suggestions for Authors

Relevant and methodologically well designed study highlighting 3MBTL3  diagnostic and prognostic potential as a biomarker in gastric cancer. Congratulations!

Comments on the Quality of English Language

Quality of English appropriate.

Reviewer 3 Report

Comments and Suggestions for Authors

This paper is a bioinformatical study looking at L3MBTL3 as a prognostic biomarker in gastric cancer. The authors are to be commended for validating their results with IHC results.

Some questions....what exactly is Figure 1B showing? Is this patient matched normal vs tumor tissue from TCGA.  Please explain how this panel is different from Fig 1 A rather than lumping them together.

In general, figure resolution is very poor, especially in figures 1 and 2.

Define the cancer type abbreviations in Figure 1A and B in the panel or in the figure legend....I am not sure which one is even gastric cancer.

Are the authors able to provide survival curves for their IHC cohort?

The Figure 2 legend should be presented in order of panel listing, not as written to fit with the sentence.  Explain each panel, and provide summary statement if needed at the end of the figure legend.

I would recommend moving the KM curves in Figure 2 to a separate figure, and define each panel explicitly.  It is not easy to follow the panels based on the current figure legend.

For the discussion, I would recommend a figure displaying a model of the proposed mechanism for L3MBTL3 function in gastric cancer.

Comments on the Quality of English Language

No major concerns, apart from rewriting the figure legends and expanding abbreviations within the legends as previously discussed.

Reviewer 4 Report

Comments and Suggestions for Authors

The aim of this study was “to investigate the association between lethal (3) malignant brain tumor-like 3 (L3MBTL3) expression and clinicopathological characteristics of gastric cancer (GC) cases, as well as its prognostic value and biological function based on large-scale databases and clinical samples”.

The paper is short, covering mainly bioinformatics and statistical studies. The results, although interesting include the role of one prognostic-diagnostic marker in GC.

The subchapter “Introduction” of the paper is written clearly, I have no special comments. The subchapter “Material and Methods” is also described in detail. I have no comments. Only in the description of “Immunohistochemistry” an error crept in on line 127 - ERAP1 expression, please correct. It should be L3MBTL3 expression, shouldn't it? In the description that the slides were viewed by 2 separate pathologists, the initials of their names should be given in parentheses (line 128).

“Results” are presented in descriptive form with citation of relevant figures and tables. The analysis is presented correctly. The results are readable and, are based on bioinformatic analysis of data available in various databases and thus enhance the value of the results obtained by the authors in the current work, especially in the context of the prognostic significance of L3MBTL3. The tables are clear and support the data well in the text description.

The discussion is brief, summarizing the results of the work and the possible benefits of the study in clinical terms. It is emphasized that L3MBTL3 plays a biological role in enhancing GC oncogenesis, and the underlying regulatory mechanisms are not fully understood and require additional basic and clinical studies. However, L3MBTL3 can be considered as a prognostic factor in terms of patient survival.

I congratulate the authors on their interestingly presented research results and wish them further work in this direction.

So this well prepared paper can be published after the additions or corrections that I have indicated.

Reviewer 5 Report

Comments and Suggestions for Authors

1. Add study design to the title AND write the abstract according to the guidelines for this study design 

2. Space should be used before references in the manuscript 

3. Line 61 - please add reference 

4. Line 71 and 73 - add normal references and cite them in the literature as internet site

5. Line 75 - which hospital

6. Figure 1 is hard to follow. Please present more appropriately 

7. Line 207 - describe all the abbreviations under the table 

8. Please add limitation section 

overall, this is an interesting study, but the manuscript could be improved in the term of English language and scientific style. Moreover, above comments could be used as a guidance.

Comments on the Quality of English Language

Could be improved 

Round 2

Reviewer 1 Report

Comments and Suggestions for Authors

The authors have made an effort to improve the manuscript, but their reasons for not including experimental data other than the immunohistochemical analyses in the revised version are not satisfactory. Although the quality of the manuscript has improved, serious objections may be still raised concerning its overall merit